

**Variations of atmospheric PAHs concentrations, sources, health risk, and direct**
**medical costs of lung cancer around the Bohai Sea under the background of**
**pollution prevention and control in China**
Wenwen Ma[1,4,5], Rong Sun[1,4,*], Xiaoping Wang[3], Zheng Zong[1,4], Shizhen Zhao[2], Zeyu Sun[1,4,5],
Chongguo Tian[1,4,*], Jianhui Tang[1,4], Song Cui[6], Jun Li[2], Gan Zhang[2]
[1] CAS Key Laboratory of Coastal Environmental Processes and Ecological Remediation, Yantai
Institute of Coastal Zone Research, Chinese Academy of Sciences, Yantai Shandong 264003, P.R.
China
[2] State Key Laboratory of Organic Geochemistry, Guangzhou Institute of Geochemistry, Chinese
Academy of Sciences, Guangzhou, 510640, China
[3] Ludong University, Yantai, 264025, China
[4] Shandong Key Laboratory of Coastal Environmental Processes, Yantai Shandong 264003, P.R.
China
[5] University of Chinese Academy of Sciences, Beijing, 100049, China
[6] International Joint Research Center for Persistent Toxic Substances (IJRC-PTS), School of Water
Conservancy and Civil Engineering, Northeast Agricultural University, Harbin 150030, China
* Correspondence to: Rong Sun (rsun@yic.ac.cn) and Chongguo Tian (cgtian@yic.ac.cn)



**Abstract**. The Bohai Sea (BS) as the most polluted area of PAHs in China has been received wide attention in recent decades. To characterize the variations of concentrations and sources of PAHs from June 2014 to May 2019, fifteen congeners of PAHs ($\sum_{15}$PAHs) were measured from atmospheric samples (N=228) collected at 12 sites around the BS, and health risk and direct medical costs associated with lung cancer exposed to PAHs were also estimated. The annual daily average concentration of $\sum_{15}$PAHs was $56.8 \pm 4.7$ ng m$^{-3}$, dominated by low molecular weight (LMW-PAHs, 3-ring) ($58.9 \pm 7.8\%$). During the five-year sampling period, the atmospheric $\sum_{15}$PAHs concentration reduced by 17.5% for the whole BS, especially in the tightly controlled area of Tianjin (TJ) with a drop of 51.7%, which was mainly due to the decrease of high molecular weight PAHs (HMW-PAHs, 5-6 ring) concentration. Generally, the concentration of $\sum_{15}$PAHs was highest in winter and lowest in summer, mainly attributed to the change of LMW-PAHs concentration. Based on PMF model, PAHs at the BS were mainly ascribed to coal combustion and biomass burning. And the contribution of coal combustion and motor vehicle to PAHs had a different performance between the BS (coal combustion rose by 6.7%, motor vehicle fell by 22.7%) and TJ (coal combustion fell by 13.2%, motor vehicle rose by 6.7%). The incidence of lung cancer (ILCR) caused by exposing to atmospheric PAHs at the BS and TJ decreased by 74.1% and 91.6% from 2014 to 2018, respectively. That was mainly due to the decrease of the concentration of highly toxic HMW-PAHs. It was reflected on the savings of $10.7 million in direct medical costs of lung cancer exposed PAHs, which was accounted 46.1% before air prevention and control around the BS. And there was a higher cost reduction of 54.5% in TJ. Hence, this study proved that implementing pollution prevention and control not only effectively reduced the concentration of pollutants and the caused risks, but also significantly reduced the medical costs of diseases caused by corresponding expose.



**1 Introduction**

Polycyclic aromatic hydrocarbons (PAHs) were a class of classical organic compounds with at least two benzene rings, and have been received long-term attention because of cytotoxic, teratogenic, mutagenic, or carcinogenic (Colvin et al., 2020; Marvin et al., 2020). The United States Environmental Protection Agency (USEPA) identified sixteen PAH congeners as priority pollutants (Lv et al., 2020). The sixteen PAH congeners in the atmosphere were considered as a major portion of lung cancer risk to the public because of their relatively high concentration, strong toxic potency, and long-term distance transmission (Ma et al., 2010; Gong et al., 2011; Ma et al., 2013; Hong et al., 2016). According to the statistics, the incidence and mortality of lung cancer were ranked first among cancer-related cases in the world, and so the lung cancer risk owing to exposing to PAHs was of particular concern and widely assessed (Jia et al., 2011; Zhuo et al., 2017; Lian et al., 2021).

PAHs were emitted primarily via incomplete combustion and pyrolysis of carbon-contained materials, such as fossil fuels and biomass (Biache et al., 2014). China has been assessed as the largest emitter of PAHs all over the world for recent two decade because of rapid development of the economy and increasing consumption of carbon-contained materials (Zhang et al., 2007). Beijing-Tianjin-Hebei (BTH) region was one of the regions with the highest PAHs emission intensity and the heaviest atmospheric PAHs concentrations in China (Zhang et al., 2007; Zhang et al., 2016). In such serious pollution, the health risk exposed to PAHs caused great concern. The population attributable fraction (PAF) for lung cancer caused by inhalation of PAHs in the atmosphere of the BTH area was more than twice higher than the mean value in whole China in 2009 (Zhang et al., 2009). The incremental lifetime cancer risk (ILCR) of the PAHs exposure at Tianjin was in the range of $1 \times 10^{-5}$ to $1 \times 10^{-3}$ in 2008, which was much higher than the mean level of $4.56 \times 10^{-6}$ in China (Lian et al., 2021; Bai et al., 2009). The annual lung cancer morbidity of Tianjin ($6.99 \times 10^{-6}$) within the BTH region was the highest city among 35 cancer registries in China (Zhang et al., 2007). Meanwhile, with the frequent occurrence of haze in the BTH region, more attention has been paid to concentration levels and health risk of fine particulate matter with aerodynamic equivalent diameter $\leq 2.5$ μm (PM$_{2.5}$) since 2013 (Chen et al., 2020).



PM$_{2.5}$ pollution in China has obviously been improved since the Air Pollution Prevention and
Control Action Plan (2013-2017) and the Three-year Action Plan for Winning the Blue-Sky
Defense Battle (2018-2020) were proposed by the Chinese government in 2013 and 2018 (Zhao et
al., 2023). As the most severely polluted area in China, the improvement was more significantly at
the BTH region, which implemented the strictest pollution control policy (Li et al., 2020). As
reported that the concentration of PM$_{2.5}$ at the BTH region dropped by 52% from 106 μg m$^{-3}$ in
2013 to 51 μg m$^{-3}$ in 2020 (Bulletin of the State of China's ecological Environment, 2021). In the
prevention and control of pollution policies, reducing emissions of coal combustion and motor
vehicle were the major parts (Guo et al., 2018; Li et al., 2019). The two sources have been
recognized as primary contributors to PAHs in the atmosphere as well (Lin et al., 2015; Han et al.,
2018). As a result, the controls of the two sources not only reduced PM$_{2.5}$ emission, but also PAHs
emission (Zhi et al., 2017). During the controlling processes, the variations in the concentrations
and health risk of PM$_{2.5}$ at BTH region have been well identified (Fang et al., 2016; Yan et al.,
2019), while the relevant understanding of PAHs in the region urgently needs to be updated.
Especially, the statistical data of the lung cancer risk due to exposing to PAHs was established ten
years ago (Zhang et al., 2009; Bai et al., 2009).
To track changes in concentrations and source of atmospheric PAHs and estimate health risk
and the direct medical costs associated with lung cancer by exposing to PAHs during the air
pollution control actions, a field monitoring campaign was conducted at twelve sites around the BS
for five years from June 2014 to May 2019. The BS is the only inland sea in China, and surrounded
by the BTH region, the Liaodong Peninsula, and the Shandong Peninsula (Liu et al., 2020). The
measures for air pollution control implemented were different at the BTH region, the Liaodong
Peninsula, and the Shandong Peninsula (Huang et al., 2017). Thus, it would provide us an
opportunity to understand the difference in environmental concentrations, source contributions, and
health risk of PAHs. The main aims of this study were (1) to characterize the spatial and temporal
changes of the concentrations and components of PAHs in the atmosphere around the BS, (2) to
evaluate the difference of source contributions of PAHs, and (3) to assess the changes of direct



medical costs for treating lung cancer caused by inhalation exposure to PAHs under atmospheric
prevention and control in the five years.
**2 Materials and methods**
**2.1 Sampling site and sample collection**
The sampling sites for this study had been reported in previous literatures (Sun et al., 2021),
and it was briefly introduced here. The information of the sites was shown in Table S1 of the
Supporting Information (SI). Twelve air sampling sites were located at Beihuangcheng (BH),
Dalian (DL), Donggang (DG), Dongying (DY), Gaizhou (GZ), Longkou (LK), Laoting (LT),
Rongcheng (RC), Tianjin (TJ), Xingcheng (XC), Yantai (YT), and Zhuanghe (ZH). A passive air
sampler with polyurethane foam (PUF, 14 cm diameter × 1.35 cm thickness) was used to collect
atmospheric samples at each sampling site (Eng et al., 2014). The PUF disks were deployed around
1.5−2.0 m above the ground, the sampling duration was about 3 months for one batch. 228 samples
were collected from June 2014 to May 2019. The sampling rate of atmospheric PAHs was 3.5 $m^3$
$day^{-1}$ (Jaward et al., 2005; Moeckel et al., 2009). Prior to sampling, the PUF disks were pre-cleaned
by methanol, acetone, and hexane, respectively. The extracted PUF disks were placed in airtight
containers and stored at −18 °C before the sampling campaign. After sampling, the samples were
prepared and then stored at a −18 °C freezer in the lab for further analyses.
**2.2 Sample pretreatment and instrumental analysis**
The five PAHs surrogates (Naphthalene-$D_8$, Acenaphthene-$D_{10}$, Phenanthrene-$D_{10}$, Chrysene-
$D_{12}$, Perylene-$D_{12}$) and the activated copper fragments were added in advance (Qu et al., 2022).
The samples were extracted for 24 h, which the elution was acetone and hexane (200mL, v:v=1:1)
through Soxhlet apparatus. The extracted solution was concentrated to 1mL with rotary evaporator
(SHB-III, Zhengzhou Greatwall Ltd., China). Then, silica-alumina column was used to obtain the
aromatic components, then the targets were obtained with 40 mL of a mixed solution of
dichloromethane and hexane (v:v=1:1). Finally, the eluent was concentrated and reduced to 500 μL
by a gentle nitrogen stream. As the internal standard substance, 400 ng of hexamethylbenzene
(Supelco, USA) was added to each sample solution before the instrumental analysis.



The targets were detected through the gas chromatograph equipped with mass spectrometry
(GC-MS, Agilent 5975C-7890A, USA), and the chromatographic column was DB-5MS (Agilent
Technologies, 30 m × 0.25 mm × 0.25 μm). Each extract was injected by 1 μL with splitless mode.
High-purity helium (purity ≥ 99.99%) with a flow rate of 1.3 mL min$^{-1}$ was used as the carrier gas.
The process of oven temperature was set as at 80 °C with a hold of 3 minutes, and then raised to
310 °C by 10 °C min$^{-1}$, and then hold 10 minutes. The temperatures of inlet and ion source were
290 °C and 230 °C, respectively. The details of the targeted compounds were shown in Table S2 of
SI. Seven gradients of mixed solutions were established for quantitative calculation of PAHs. More
details were reported in previous study (Wang et al., 2018).
**2.3 Quality assurance and quality control**
The mean recovery values of Naphthalene-$D_8$, Acenaphthene-$D_{10}$, Phenanthrene-$D_{10}$, Chrysene-
$D_{12}$, and Perylene-$D_{12}$ were 77.3%, 85.9%, 87.5%, 88.3%, and 92.8%, respectively, which were
ranging from 66.5% to 123.1%. All the relative deviations were within 20%, except for
Naphthalene-$D_8$. Nap was excluded because of its low recovery, and the other fifteen PAHs
($\Sigma_{15}$PAHs) were used for further discussion in this study. For each batch of twelve PUF samples, a
field blank and a procedural blank were also analyzed at same treatment process. In this study, the
method detection Limits (MDLs, defined as the mean blank value plus 3 times the standard
deviation) for 15 PAH congeners ranged from 0.016 to 0.126 ng sample$^{-1}$, which were shown in
Table S2 of SI. The final concentrations were not surrogate-corrected. The glassware was all
cleaned and burned for 8 hours in muffle oven at 450 °C before the experiment. The solvents were
chromatography-pure or had been redistilled and purified before using.
**2.4 Source apportionment of PAHs**
The model of positive matrix factorization (PMF) released by the USEPA (PMF 5.0) was used
to apportion the emission sources of PAHs in this study. The basic calculation formula of the PMF
method is as Eq. (1):
$$x_{ij} = \sum_{k=1}^{p} g_{ik} f_{kj} + e_{ij} \tag{1}$$





where p represents the number of sources identified by the PMF model. $x_{ij}$ represents original
concentration data of $i^{th}$ chemical species and $j^{th}$ sample. $f_{ik}$ represents the source profile of $k^{th}$
source and $j^{th}$ chemical species. $g_{kj}$ represents contribution ratio of $k^{th}$ source to $j^{th}$ sample. $e_{ij}$
represents the simulated residual error of $i^{th}$ chemical species and $j^{th}$ sample. Source contributions
and profiles are solved by the PMF model minimizing the objective function $Q$, as Eq. (2):
$$Q_{\min} = \sum_{i=1}^{n}\sum_{j=1}^{m}\left(\frac{x_{ij}-\sum_{k=1}^{p}g_{ik}f_{kj}}{u_{ij}}\right)^2 \qquad (2)$$

where $x_{ij}$, $g_{ik}$, and $f_{kj}$ are same that in Eq. (1), respectively. $u_{ij}$ is the uncertainty of $x_{ij}$, and the
calculation method of uncertainty is showed in Text S2 of SI. More details have been documented
(Sofowote et al., 2011; Paatero et al., 2014).
Before the source apportionment, principal component analysis (PCA) was applied to pre-
estimate the minimum number of emission sources in this study because PCA was able to explain
the overall variables with fewer variables with a minimum loss of information (Liu et al., 2021).
SPSS Statistics 25.0 was used to perform the PCA analysis in this study.
**2.5 Health risk assessment**
The total toxicity equivalent (*TEQ*, ng m$^{-3}$) of the fifteen PAHs with *BaP* as reference is
calculated as Eq. (3):
$$TEQ = \sum_{i=1}^{n}(C_i \times TEF_i) \qquad (3)$$

where $C_i$ is concentration of the $i^{th}$ PAH compound (ng m$^{-3}$), $TEF_i$ is the cancer potency of the
$i^{th}$ PAH compound (dimensionless), as shown in Table S2 of SI.
*ILCR* in this study referred to cancer risk in a population due to exposure to a specific
carcinogen (Zhuo et al., 2017). Its calculation formula is as Eq. (4):
$$ILCR = UR_{BaP} \times TEQ \qquad (4)$$

In above, $UR_{BaP}$ represents the cancer risk when the concentration of *BaP* is 1 ng m$^{-3}$ (ng m$^{-}$
$^3$). According to the regulations of World Health Organization (WHO), $UR_{BaP}$ can be $8.7 \times 10^{-5}$ per



ng m$^{-3}$. That is, in terms of life span of 70 years, lifetime exposure to $BaP$ concentration of 1 ng m$^{-}$
$^3$ resulted in a risk of cancer by inhalation of $8.7 \times 10^{-5}$ (Luo et al., 2021).
**2.6 Medical costs assessment**

180       In this study, the medical costs were assessed by comparing total direct medical costs for

treating lung cancer caused by respiratory exposed to PAHs in the atmosphere under the assumption
that no air pollution control and the actual implementation of air pollution control. The total direct
medical costs for treating lung cancer ($C_t$) are calculated as Eq. (5):
$$C_t = C_{pc} \times P \times I_{add} \tag{5}$$

185       where $C_t$ is the total direct medical costs of lung cancer induced by PAHs exposure, $C_{pc}$ is the

per capita direct medical costs of lung cancer, and a cost of \$8,700 in China in 2014 was used in
this study (Shi et al., 2017). $P$ is the annual population, $I_{add}$ is the additional incidence of lung cancer
due to PAHs inhalation exposure, it is calculated as Eq. (6):
$$I_{add} = I \times PAF \tag{6}$$

190       where $I$ is the incidence of lung cancer. And the $I$ value was $87.37 \times 10^{-5}$ at Tianjin estimated

in 2012, which was referred in this study (Cao et al., 2016). $PAF$ is the population attributable
fraction, defined as the decrease in the incidence or mortality of a disease when a certain risk factor
is completely removed or reduced to another lower reference level (Menzler et al., 2008). The $PAF$
can be calculated as Eq. (7):
$$PAF = \frac{rr(TEQ) - 1}{rr(TEQ)} \quad \text{and} \quad rr(TEQ) = [URR_{cum,\ exp\ =\ 100}]^{(TEQ \times 70/100)} \tag{7}$$

196       where $rr$ is relative risk, that is, the risk of exposure to a specific concentration relative to no

exposure. $URR$ is the unit relative risk, a reference value of 4.49 per 100 μg m$^{-3}$ years of $BaP$
exposure was adopted in this study (Zhang et al., 2009). This reference value was based on an
epidemiological study on lung cancer conducted in Xuanwei, China (Menzler et al., 2008) (Gibbs
et al., 1997). This study assumed that the mean life expectancy in China was 70 years, and the
lifetime exposure was equivalent to 70 years.
**3 Results and discussions**





### 3.1 Concentration and composition of PAHs

#### 3.1.1 General information of PAHs

Figure 1 summarizes the annual daily average concentrations of 15 PAHs in the atmosphere at the twelve sampling sites around the BS from June 2014 to May 2019. The annual daily average concentration of $\Sigma_{15}$ PAHs around the BS was $56.8 \pm 4.7$ ng m$^{-3}$, with a range of $51.4 - 63.6$ ng m$^{-3}$. And the highest concentration was the low molecular weight PAHs (LMW-PAHs, 3-ring), followed by middle molecular weight PAHs (MMW-PAHs, 4-ring) and high molecular weight PAHs (HMW-PAHs, 5-ring and 6-ring), which were accounting for 58.7%, 34.8%, and 6.65% of the total concentration, respectively. The atmospheric PAHs concentration was dominated by the LMW-PAHs in this study, which Phe, Fla, and Flu were the main compounds accounting for 37.7%, 19.8%, and 12.6% of the total. The atmospheric PAHs concentrations around the BS were at a higher pollution level than the Yangtze River Delta and the Pearl River Delta, such as Ningbo (45 ng m$^{-3}$) (Tong et al., 2019) and Guangzhou (9.72 ng m$^{-3}$) (Yu et al., 2016). And the atmospheric concentrations of PAHs around the BS were also much higher than in atmosphere above the Great Lakes (1.3 ng m$^{-3}$) (Li et al., 2021) and southern Europe cities (3.1 ng m$^{-3}$) (Alves et al., 2017). Overall, it was found that the pollution of atmospheric PAHs around the BS was still worrying.

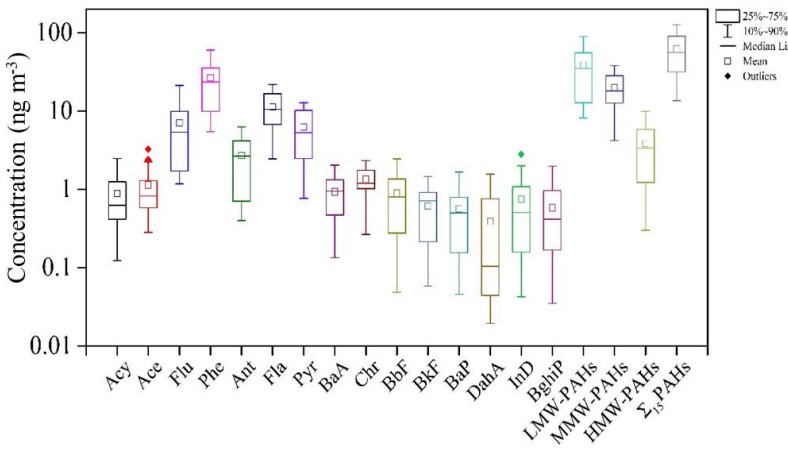



**Figure 1**. Atmospheric concentrations of PAHs around the BS from June 2014 to May 2019.

**3.1.2 Temporal variations of PAHs**
For seeking better understand the variation characteristics of PAHs in the atmosphere, the
summer of the previous year to the spring of the next year were taken as a statistical cycle. The
concentrations of $\Sigma_{15}$PAHs in the five annual cycles around the BS were $63.6 \pm 58.4$ ng m$^{-3}$ (2014-
2015), $55.5 \pm 37.9$ ng m$^{-3}$ (2015-2016), $60.9 \pm 31.1$ ng m$^{-3}$, (2016-2017), $51.4 \pm 29.4$ ng m$^{-3}$ (2017-
2018), and $52.5 \pm 40.1$ ng m$^{-3}$ (2018-2019), respectively (Table S3 of SI). Overall, the
concentrations of $\Sigma_{15}$PAHs from June 2014 to May 2019 showed a slow downward trend with a
decrease of 17.5%. The decrease of atmospheric PAHs concentrations was mainly due to the
decline of the HMW-PAHs concentrations. The HMW-PAHs composition ratio decreased from
11.3% (2014-2015) to 3.44% (2018-2019), while the MMW-PAHs raised from 35.5% (2014-2015)
to 41.2% (2018-2019). The LMW-PAHs composition ratio was stable from 53.4% (2014-2015) to
55.4% (2018-2019). The one factor that effected the concentrations of PAHs in the atmosphere
after they were discharged from the emission source was meteorological conditions (Fan et al.,
2021), and the other important factor was the amount of the direct emission from the emission
source (Ma et al., 2018). The sources of PAHs with different ring numbers were different (Li et al.,
2021). LMW-PAHs were mainly produced in the combustion process of non-petroleum sources,
while HMW-PAHs were mainly from high temperature combustion products generated by fossil
fuel combustion, including some activities involving pyrolysis process, such as vehicle emissions,
industrial productions, and other high-temperature source emissions (Zhang et al., 2018; Xing et
al., 2020). The significant decrease of HMW-PAHs concentrations at the BS during the five-year
observation period might be related to the decrease of high temperature emission sources. Due to
the high toxicity characteristics of HMW-PAHs (Biache et al., 2014; Ma et al., 2020), the decrease
of its concentration might indicate a decrease in the environmental toxicity of PAHs.
The seasonal distributions of PAHs concentrations in the atmosphere of the BS region showed
high in cold season and low in warm season. The concentrations of $\Sigma_{15}$PAHs in four seasons were



as follow: winter ($104.3 \pm 9.50$ ng m$^{-3}$) > autumn ($53.9 \pm 9.10$ ng m$^{-3}$) > spring ($43.9 \pm 19.5$ ng m$^{-3}$) > summer ($26.3 \pm 13.4$ ng m$^{-3}$) (Table S5 of SI). The concentration of PAHs in winter was about
4 times higher than that in summer, and the maximum and minimum of the annual daily average
concentrations at 12 sampling point mostly occurred in winter and summer. In addition, there were
significant differences between total PAHs concentration and different ring number concentrations
($p < 0.05$, the difference level is shown in Table S6 of SI). The seasonal characteristics of PAHs
concentrations in this study were consistent with reported results in North China (Ma et al., 2017;
Zhang et al., 2019). Interestingly, it was that the difference of PAHs concentrations in four seasons
was mainly on account of LMW-PAHs. This indicated that there were other important pollution
sources for LMW-PAHs, followed by MMW-PAHs, which was significantly increasing in winter
at the BS region. Then identifying the source of LMW-PAHs was crucial for improving
environmental quality of the BS. Studies have shown that coal burning emissions and biomass
burning were the main sources of atmospheric PAHs in this region (Liu et al., 2019). For typical
northern families, the consumption of firewood burning and coal in winter was 1.5−2.0 times higher
than that in summer due to heating and other activities (Qin et al., 2007). As a result, PAHs
emissions in winter were at least 1.5 times higher than those in summer. In addition, due to the
migration characteristics of atmospheric PAHs, meteorological conditions such as temperature and
wind direction in different seasons would also affect the observed concentration (Tan et al., 2006).
And low temperature and inversion layer in winter were not conducive to atmospheric diffusion,
resulting in a relatively high concentration of PAHs in the atmosphere near the surface (Wang et
al., 2018).

### 3.1.3 Spatial characteristics of PAHs

Figure 2 and Table S7 of SI displays the distribution of the five-year mean concentrations of $\Sigma_{15}$
PAHs from June 2014 to May 2019 at the 12 sampling sites around the BS. The concentrations of
atmospheric $\Sigma_{15}$PAHs ranged from $25.9 \pm 6.4$ ng m$^{-3}$ (RC) to $103.7 \pm 39.1$ ng m$^{-3}$ (XC). The
concentrations of PAHs on the BS north coast were twice higher than at the BS south coast. PAHs
were a class of pollutants that can undergo long-range transport in the atmosphere (Wang et al.,





2018), and their spread was largely affected by local meteorological conditions (Ding et al., 2005).
The climate in North China and the adjacent oceanic area was greatly affected by the East Asian
monsoon, and the characteristic weather phenomenon in the winter monsoon was the strong north
and northwest winds (Tian et al., 2009). Due to the additional emissions from fuel consumption for
domestic heating in the source areas, the atmospheric PAHs concentrations significantly increased
(Feng et al., 2007; Gao et al., 2016). Combined with backward trajectory shown in Fig. S4 of SI, it
suggested that the elevated PAH concentrations in winter at the north of the BS were mainly
attributed to their outflow from the north and northwest source regions carried by the winter
monsoon winds. However, the composition of PAHs at the north-south showed consistency without
no significant differences (Table S8 of SI). As the whole, the composition of PAHs at 12 station
that the highest content was LMW-PAHs (North: 60.0%, South: 57.4%), followed by MMW-PAHs
(North: 32.7%, South: 32.4%), and HMW-PAHs was the lowest (North: 7.3%, South: 10.8%). The
above indicated that there were the same emission sources of PAHs in the atmosphere around the
BS.

288        However, for TJ, the study found that there was a more significant change in the concentration

of atmospheric PAHs, which decreased from 68.6 ng m$^{-3}$ (2014-2015) to 33.1 ng m$^{-3}$ (2018-2019).
The reason was mainly that TJ was located at the Beijing-Tianjin-Hebei region where was the
strictest area of air pollution prevention and control, as a key area in China's "12th Five Year Plan".
For exploring the potential differences of source emissions at 12 sampling points, Pearson
correlation analysis was used to analyze the seasonal distribution of PAHs concentrations as shown
in Table S9 of SI. Among the five stations (LK, DY, TJ, LT, and XC) at the western BS centered on
TJ, the correlation coefficients of atmospheric PAHs concentration (0.72–0.89) among the other
four stations were greater than that between each site and TJ (0.50–0.68). That the co-variability
of PAHs concentrations between TJ and the other four stations was weaker. This indicated that
there were certain differences between TJ's PAHs emission sources and adjacent areas.




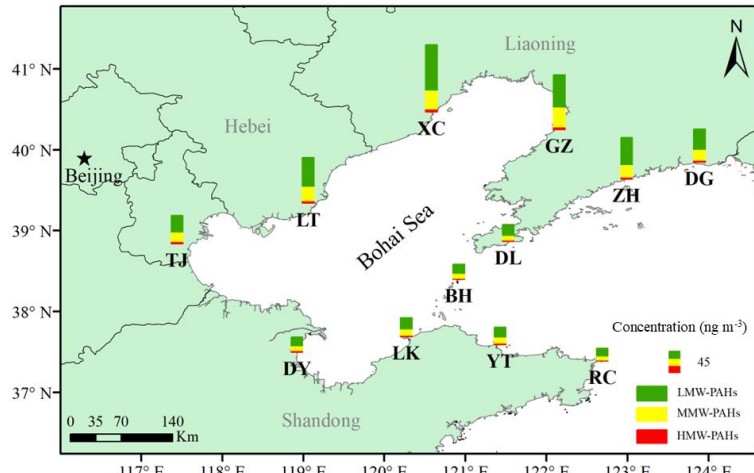

**Figure 2**. The mean concentration distribution of $\Sigma_{15}$ PAHs at 12 sites around the BS from June 2014 to May 2019.


**3.2 Source apportionment of PAHs**

For further probing into the causes for the variations of the concentrations and compositions of PAHs, the sources apportionment of PAHs in the atmosphere around the BS and TJ region from 2014-2015 to 2018-2019 was investigated via PCA and PMF. PCA analysis results showed that when four factors (eigenvalues > 1) were extracted from the data set, the total cumulative load accounted for more than 85% of the variance (Table S10 of SI). This indicated that at least four types of emission sources could better explain the source of atmospheric PAHs. For PMF model, the key process was to determine the correct number of factors, and this study was based on the results of PCA. Based on the random seed, 4 − 7 factors were used through the PMF model for source analytical simulation. The source analytical simulation of five factors determined the most stable results and the most easily interpreted factors. The solution produces $Q$ values (both robust and true) that were close to the theoretical $Q$ values, which was indicating that the PAHs data set in the modeling input provided appropriate uncertainty. The data set used for PMF analysis



included the concentrations of 228 samples of 15 PAHs and uncertainties. The diagnostic regression
$R^2$ value for the overall concentrations of 15 PAHs components was 0.986. The predicted
concentrations of 15 PAHs via PMF model were almost consistent with the actual concentrations
of 15 PAHs around the BS (Fig. S5−S6 of SI and Text S2 of SI). It meant that the model results
were good and could be used as the judgment basis for source analysis of target species, so these 5
factors would well explain the source of PAHs. Contribution of source identified by PCA and PMF
were coal combustion, biomass burning, industrial processes, gasoline emission, and diesel
emission. The detailed information of source identification is shown Text S3 of SI.

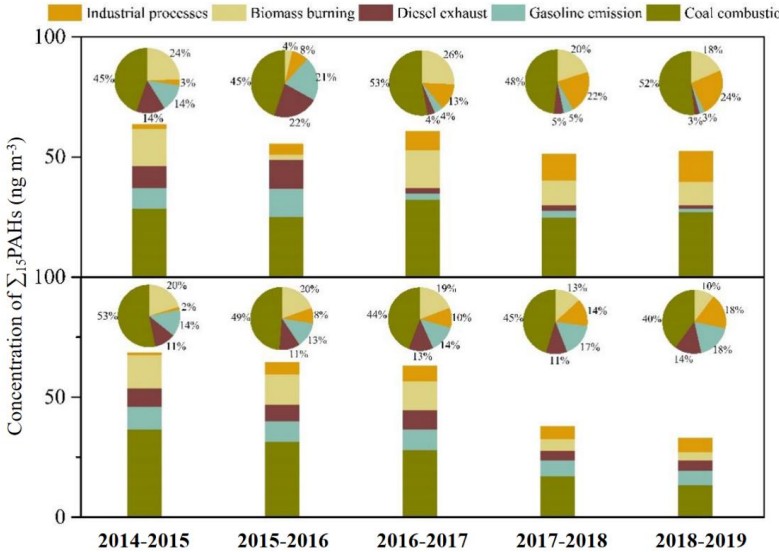

**Figure 3**. Concentration and source contribution of $\Sigma_{15}$ PAHs sources around the BS (the upper
part) and TJ (the lower part) from 2014-2015 to 2018-2019.

Fuel combustion emissions were the reason for the significant increase of atmospheric pollutants,
and that were also responsible for the elevated $PM_{2.5}$ levels around the BS region (Yang et al., 2017).
To explore the relationship between $\Sigma_{15}$PAHs and $PM_{2.5}$ concentrations, available online $PM_{2.5}$ data



for eight cities that on behalf of sampling sites (DG, DL, DY, GZ, LT, TJ, XC, and YT) (Air quality
historical data query, 2014-2019) were collected, which averaged their concentrations according to
the sampling periods in the study (Table S12 of SI). The Pearson correlation coefficients of the
concentrations of atmospheric PAHs and $PM_{2.5}$ were ranging from 0.485 to 0.868, and the
significant levels were greater than 95% as listed in Table S13 of SI. During the observation of the
five-year, the $PM_{2.5}$ concentration at the BS region decreased by 29.6% from 57 μg m$^{-3}$ to 40 μg m$^{-3}$,
and at TJ showed an even greater decrease by 33.8% from 78 μg m$^{-3}$ to 51 μg m$^{-3}$. From 2013,
$PM_{2.5}$ had been strictly controlled by the government year by year, which the significant correlation
indicated that the PAHs concentrations should be affected. To explore the potential influencing
factors of the difference in atmospheric PAHs composition between the BS area and TJ, their
average annual contributions of various PAHs emission sources from 2014-2015 to 2018-2019
were compared shown in Figure 3. During the sampling period of the BS region, coal combustion
was the main source of the atmospheric PAHs emission (45%), followed by biomass burning (24%)
in 2014-2015, which was switching to coal combustion (52%) and industrial processes (24%) in
2018-2019. For TJ, coal combustion was also the main source of the atmospheric PAHs emissions
(53%), followed by biomass burning (20%) in 2014-2015, which was switching to coal combustion
(40%), industrial processes (18%) and gasoline emissions (18%) in 2018-2019. The source
contributions of coal combustion to atmospheric PAHs had increased by 7% around the BS, while
the corresponding contributions in TJ had fallen by 13%. The absolute contribution (the total
concentration of PAHs multiplied by the percentage value of the contributing source) decreased,
which was indicating that the reduction of the coal contribution source had a significant
improvement on the atmospheric PAHs pollution.
The main source of atmospheric PAHs around the BS was coal combustion (Liu et al., 2019; Qu
et al., 2022), while for TJ, as one of the key areas for air pollution control in China, had taken
stricter measures to control emissions of coal combustion (Wu et al., 2015). For instance, the city
took the lead in the switching domestic fuel from coal to natural gas and electricity in 2017 to
reduce emissions of air pollutants (Zhang et al., 2021). These targeted measures had more



forcefully controlled coal-combustion emissions for PAHs in TJ than the other places around the
BS region (Guo et al., 2018). Vehicle emission (gasoline and diesel exhaust) to atmospheric PAHs
had experienced a sharp drop of 23% for the BS area, while for TJ risen by 7%. The same trend for
vehicle emission was found in the study of Beijing and Tianjin (Zhang et al., 2016; Chao et al.,
2019). The decrease was mainly due to the elimination and scrapping of substandard vehicles
carried out by the Chinese government in 2015. Based on the "China Vehicle Environmental
Management Annual Report", the car ownership around the BS increased by about 17.5 million,
but the emissions of hydrocarbons including PAHs reduced by 95,000 tons from 2014 to 2018 (Fig.
S8 of SI). The source apportionment showed that the contribution of vehicle emission to PAHs had
a sharp decline since the spring of 2016 (Fig. S9 of SI), with a decreased by 38% (19% for gasoline
and 19% for diesel) around the BS (Huang et al., 2017). Although the contribution of vehicle
emissions for TJ was increased, the concentrations of PAHs was decreasing. It indicated that these
measures had also controlled vehicle emissions and kept the emissions of PAHs at a low level.
Therefore, targeted control measures could effectively control $PM_{2.5}$ and PAHs pollution in the
atmosphere at the BS and TJ. Moreover, PAHs were a kind of organic compounds produced with
black carbon (BC), and, to some extent, the molecular characteristics of PAHs also provided the
basic data to analysis of the source of BC in the atmosphere of the BS (Fang et al., 2016). At the
same time, the PAHs source analysis results of this study revealed that the composition and source
of atmospheric BC in the BS region have also changed from 2014 to 2019. This problem needs our
attention and confirmation.





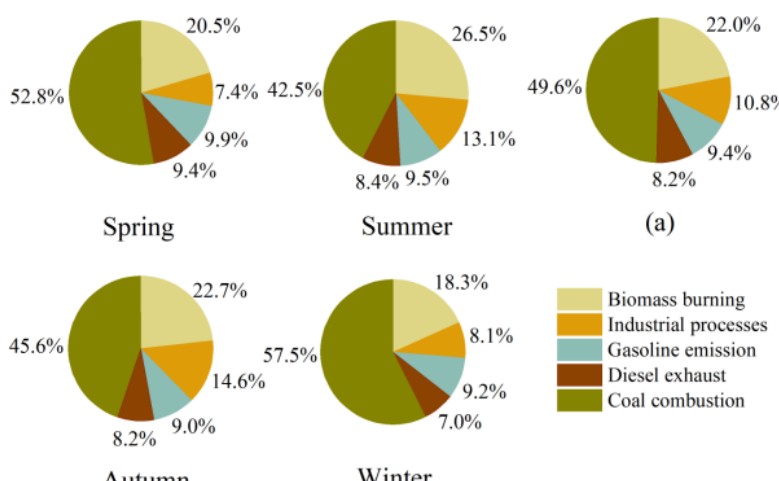

**Figure 4.** The seasonal and average contributions for five sources of $\Sigma_{15}$ PAHs derived from PMF;
(a): the five-year average contributions of five sources.

Figure 4 shows the seasonal distribution of five sources for atmospheric PAHs at the BS. Generally, the seasonal distribution of five sources for atmospheric PAHs at TJ was consistent with that the BS, which was not separately discussed here. Coal combustion was the main emission source in the four seasons, followed by biomass burning, while the contributions of the others (industrial processes, gasoline emission, and diesel emission) were similar. Compared with other seasons, the contribution of coal combustion for atmospheric PAHs to the BS was the highest in winter, which was followed by spring, and the lowest was in summer. This was consistent with the seasonal distributions of the concentrations of PAHs in the atmosphere at the BS. Based on the seasonal distribution of concentrations, the increase concentrations of atmospheric PAHs in winter were mainly caused by coal combustion. This might be due to people in cold winters at northern China rely on coal combustion for heating. For biomass combustion, it was higher in summer and autumn, which was related to straw burning after harvest. Given all this, the seasonal distributions of PAHs sources indicated that the pollution of atmospheric PAHs was mainly influenced by human activities.



**3.3 Health risk exposed to PAHs**


On the basis of the Eq. (3), the annual mean *TEQ* value around the BS was 1.37 ± 1.05 ng m⁻
³ from June 2014 to May 2019, which below the national standard (10 ng m⁻³) while slightly higher
than the WHO standard (1 ng m⁻³). The HMW-PAHs contributed dominantly 76.4% of the total
*TEQ*. However, the concentration of HMW-PAHs in the atmosphere accounted for 6.5% of the total
PAH concentration. Among which, the two major *TEQ* contributors were BaP (38.2% ± 8.0%) and
DahA (16.6% ± 9.0%). For TJ, the annual mean *TEQ* value was 1.69 ± 1.50 ng m⁻³, which was
slightly higher than that the BS. It was indicating that higher health risk was caused by PAHs
exposed at TJ than around the BS. The HMW-PAHs contributed dominantly 90.9% of the total
*TEQ*. However, the concentration of HMW-PAHs in the atmosphere accounted for 8% of total
PAHs concentrations. Among which, the two major contributors were BaP (47.2% ± 9.2%) and
DahA (19.7% ± 16.2%).
The information of *TEQ* at BS and TJ from June 2014 to May 2019 was shown in Figure 5.
The average value of *TEQ* at the BS in the five cycle years was 2.55 ± 1.49 ng m⁻³, 2.49 ± 1.63 ng
m⁻³, 0.69 ± 0.76 ng m⁻³, 0.47 ± 0.66 ng m⁻³, and 0.67 ± 0.84 ng m⁻³, respectively. The value of *TEQ*
at the BS showed a downward trend year by year. The decrease in the fifth year compared with the
first year was more than two times, indicating that the environmental health risk of PAHs was
decreasing. It was found that the decrease of HMW-PAHs concentration was the main reason for
the decrease of the toxicity of PAHs. For example, the concentration of BaP in the atmosphere at
the BS decreased by 79.1% in five years, and the concentration of DahA, as a species with
carcinogenic toxicity equivalent to BaP, decreased by 96.1%. For TJ, the average value of *TEQ* in
the five cycle years was 3.63 ± 0.14 ng m⁻³, 3.38 ± 0.72 ng m⁻³, 0.84 ± 0.38 ng m⁻³, 0.28 ± 0.10 ng
m⁻³, and 0.31 ± 0.15 ng m⁻³, respectively. The *TEQ* value of PAHs in the atmosphere decreased by
91.5% at TJ in the past five years. At TJ, BaP and DahA as the major contributing factors of *TEQ*
in the atmosphere also showed more significant decline than around the BS. To sum up, the results
showed that pollution control could not only reduce the total concentration of PAHs in the
atmosphere at the BS, but also affected the composition of the PAHs. And it mainly affected the



423 concentration of HMW-PAHs compounds, which the total toxic equivalent of PAHs in the

424 atmosphere at the BS was remarkably reduced.


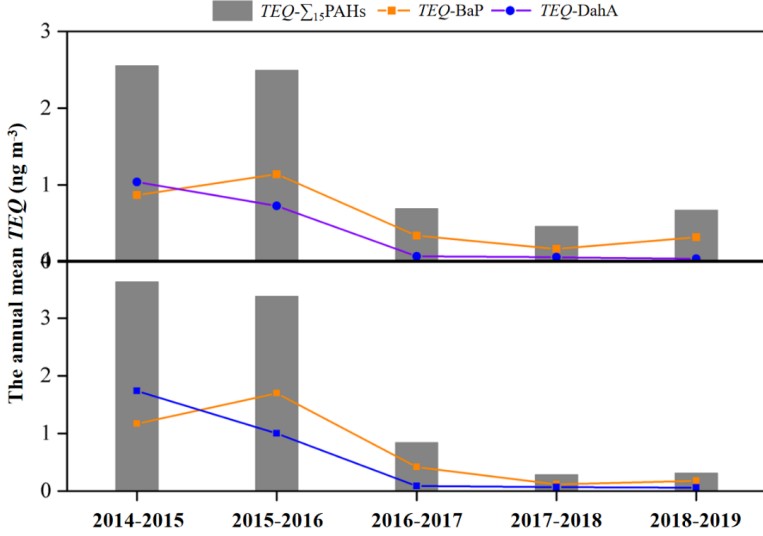

426 **Figure 5.** The annual mean *TEQ* of 15 PAHs, BaP, and DahA in the atmosphere at the BS (the

427 upper part) and TJ (the lower part) from June 2014 to May 2019.


429 Simultaneously, incremental lifetime cancer risk (*ILCR*) was used to assess the potential

430 carcinogenic risk of PAHs in the atmosphere at the BS. According to the USEPA, the *ILCR* value

431 less than $1 \times 10^{-6}$ was an acceptable risk level. When the *ILCR* value was equal to or high than $1 \times$

432 $10^{-6}$ but less than $1 \times 10^{-4}$, which was in a serious risk of cancer, and health issues should be taken

433 seriously. When the *ILCR* value were equal to or greater than $1 \times 10^{-4}$, it was considered life-

434 threatening for human. The specific calculation was seen Eq. (4). It was found that the range of

435 *ILCR* value of atmospheric PAHs at the BS region for five years was $4.1 \times 10^{-5}$–$2.2 \times 10^{-4}$, with an

436 average value of $1.2 \times 10^{-4}$, which means that the risk of cancer in this region was in a serious state,

437 and health problems should be paid more attention to. Similarly, to the above *TEQ*, *ILCR* values

438 were also dominated by HMW-PAHs. The *ILCR* caused by PAHs is listed in Table S14 of SI. The



*ILCR* at the BS decreased significantly by 74.1% from $2.2 \times 10^{-4}$ in the first year to $5.7 \times 10^{-5}$ in
the fifth year. Compared with the BS, the ILCR at TJ decreased more significantly, from $3.2 \times 10^{-4}$
to $2.7 \times 10^{-5}$ by 91.6%. As shown in Table S15 of SI, the study found that the concentration
variations of highly toxic BaP and DahA were basically synchronized with the changes of *ILCR*,
which implied that the decrease of concentrations of both was the main reason for the cancer risk
reduction. The significant reduction of cancer risk at the BS region indicated that the emission of
highly toxic HMW-PAHs in the atmosphere has been effectively controlled, which also reflected
that the prevention and control of air pollution had effectively reduced the health risk. In particular,
the reduction effect of PAHs exposure risk was more obvious at TJ, which air pollution control was
strict.

### 3.4. Direct medical costs of lung cancer caused by exposed to PAHs

This reduction of PAHs health risk would lead to a reduction in the number of people who
develop cancer, thus saving on the cost of cancer treatment. In this study, the direct medical costs
of lung cancer caused by respiratory exposure to PAHs was estimated by the additional incidence
of lung cancer caused by PAHs exposure, the population in the study area, and the direct medical
costs per capita of lung cancer patients. The specific calculation was seen Eq. (5). In addition to
PAHs exposure, there were many environmental risk factors that could induce lung cancer. For
deriving the lung cancer burden caused by atmospheric PAHs respiratory exposure from the
incidence of lung cancer, this study was characterized by percentage of population risk attribution
(*PAF*). The details were seen Eq. (6) and Eq. (7). *PAF* here represented the percentage of reduction
in lung cancer incidence which PAHs, an environmental factor, were completely removed or their
concentration was reduced. According to the above introduction of *PAF* and analysis of *TEQ*, the
directly calculated *PAF* within five years around the BS ranged from 0.5‰ to 2.7‰, with an
average value of 1.4‰. The five-year *PAF* at TJ ranged from 0.3‰ to 3.8‰, with an average value
of 1.7‰. A remarkable situation was that *PAF* around the BS region and TJ decreased significantly
in the past five years, from 3.8‰ and 2.7‰ in the first year to 0.3‰ and 0.7‰ in the fifth year
respectively. The additional lung cancer incidence ($I_{add}$) due to respiratory exposure to PAHs was



calculated using the product of lung cancer incidence and *PAF*. Previous studies reported that the
incidence of lung cancer at TJ in 2012 was $87.37 \times 10^{-5}$ (Cao et al., 2016). In this study, $87.37 \times$
$10^{-5}$ was used as the reference value of lung cancer incidence. The average $I_{add}$ caused by
respiratory exposure to PAHs around the BS region and TJ were $1.26 \times 10^{-6}$ and $1.55 \times 10^{-6}$,
respectively. During the observation of the five-year, the $I_{add}$ around the BS region and TJ decreased
from $2.34 \times 10^{-6}$ and $3.33 \times 10^{-6}$ in the first year to $6.15 \times 10^{-7}$ and $2.87 \times 10^{-7}$ in the fifth year,
respectively. The population numbers in the study area were all referred from the public data of the
statistical yearbook. The estimated results of the BS region and TJ are shown in Table S16–S17 of
SI, respectively. It had been reported that the direct cost of an average case of lung cancer patients
in China in 2014 was \$9042.79 (Shi et al., 2017; Huang et al., 2016). Since there was no reference
data available for other corresponding years, this study took the direct cost per case of lung cancer
patients as the baseline in 2014, and the estimate assumed the same direct medical costs per capita
for lung cancer within five years.

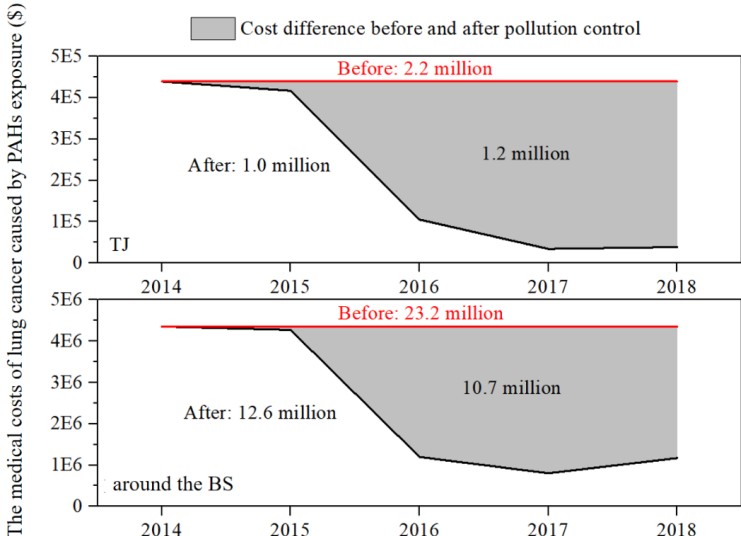

**Figure 6.** The medical costs of lung cancer caused by PAHs exposure before and after the control
of air pollution at TJ and around the BS from 2014 to 2018.




Figure 6 shows the comparative results of direct medical costs of lung cancer at the BS region
and TJ under two scenarios from 2014 to 2018. In the five years, under the implementation of air
pollution control, the total direct medical costs of lung cancer caused by respiratory exposure to
PAHs in the Bohai Rim region was $12.6 million. Assuming that no air pollution control was
implemented, the total direct medical costs of lung cancer caused by PAHs exposure did not change
in five years, and the total direct medical costs was five times in 2014 with an estimated value of
$23.2 million. The actual implementation of control on the total direct medical costs of lung cancer
saved $10.7 million. At TJ, the total direct medical costs of lung cancer induced by respiratory
exposure to PAHs under actual air pollution control was $1.0 million. Under the assumption that
no air pollution control was implemented, the total direct medical costs of lung cancer caused by
PAHs exposure was $2.2 million, saving about $1.2 million at TJ. Compared to without air
pollution control, the total direct medical costs of lung cancer caused by PAHs exposure decreased
by 46.1% around the BS region and by an even greater 54.5% at TJ. This illustrated that the
implementation of air pollution control not only reduced the risk of lung cancer caused by PAHs
exposure around the BS region, but also created significant health benefit in the direct medical
costs of lung cancer, especially in tightly controlled areas such as TJ. Therefore, the above results
noted that more precise pollution prevention and control could better reduce the emission of the
pollutants, and sequentially reduce the health risk of human expose.

**4 Conclusions**
A five-year atmospheric PAHs observation was conducted at twelve sites around the BS from
June 2014 to May 2019. The five-year atmospheric concentration of $\Sigma_{15}$PAHs was $56.8 \pm 8.4$ ng m$^-$
$^3$, characterized by dominant LMW-PAHs ($58.7 \pm 7.8\%$). The maximum annual concentrations and
seasonal concentrations occurred in the first year and every winter, respectively. The concentrations
of $\sum_{15}$ PAHs in the atmosphere reduced significantly around the BS, especially at the sampling site
of TJ during the sampling period. The contributions of coal combustion and vehicle emission to



PAHs in the atmosphere during the sampling period showed an increase and a decrease around the
BS, respectively. However, the variations of coal combustion and vehicle emission in the source
contributions in TJ were just the opposite. From 2014 to 2018, the additional lung cancer incidence
of lung cancer caused by PAH exposure around the BS dropped by 74.1%, and a higher drop of
91.6% in TJ. From the statistical standpoint, the drop of the incidence saved about $10.7 million
for the total direct medical costs of lung cancer caused by PAHs exposure around the BS. Compared
to without air pollution control, the total direct medical costs of lung cancer caused by PAHs
exposure decreased by 46.1% around the BS region and by an even greater 54.5% at TJ. And it was
further be certified that pollution reduction was beneficial to human health. In the fight against air
pollution, more precise pollution prevention and control strategies were needed.

**Data availability.** Corresponding data for the samples can be accessed on request to the
corresponding author (Chongguo Tian, cgtian@yic.ac.cn)

**Author contributions.** CT and ZZ designed the research; WM, RS, XW, ZZ, ZS, and CT
conducted the sample collection; WM, RS, and XW performed the chemical analyses; WM, RS,
XW, and CT analyzed the data, carried out the simulations and wrote the original article; ZS, SZ,
JT, SC, JL, and GZ helped with article submissions. All authors have given approval to the final
version of the manuscript.

**Competing interests**. The contact author has declared that none of the authors has any competing
interests.

**Acknowledgements.** This study was supported by the National Natural Science Foundation of
China (No. 41977190 and 42177089).

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
