# Peer review of "Variations of atmospheric PAHs concentrations, sources, health risk, and direct"

_EGUsphere, 2023_

## Author Comment (AC1)

The study by Ma et al. (Variations of atmospheric PAHs concentrations, sources, health risk, and direct medical costs of lung cancer around the Bohai Sea under the background of pollution prevention and control in China) conducted 5-years observation of polycyclic aromatic hydrocarbons (PAHs) at 12 sampling sites at the coastal area of Bohai Sea. The data were analyzed using the positive matrix factorization (PMF) for source apportionment. The major findings include reduction of PAHs during the observation period that is likely related with implementation of emission control policy and identification of coal combustion as a major emission source of PAHs in the region. The technical approach of the study sounds. It is a useful dataset, and the topic is suitable for the journal. Descriptions in the manuscript are redundant at many places. The reviewer believes that the manuscript will be much easier to be read by employing short and direct expressions. The reviewer suggests the authors to ask for a professional editorial service for better organizing the manuscript, and correct grammatical errors.

Thank you very much for the suggestion. We have asked a native English speaker to polish the grammar of the manuscript, such as Line 32, Line 54-55, Line 100, Line 262-263, and Line 493-494.

Comments

(1) Abstract: Acronyms (e.g., PAHs) should be defined before its first appearance.

Thanks for the suggestion. It has been modified, see Line 22-23.

(2) References for introduction

This is a study for atmospheric observation in China. It is not surprising that the introduction section contains numerous papers by research groups in China. However, the reviewer feels that the fraction of references from China is too high so that it could hinder contributions of researchers in other countries to the area. For instance, health risks of PAHs have widely been investigated all over the world. The reviewer suggests the authors to re-conduct literature survey and reorganize the introduction. The revision will help attracting attention of scientists in the area from other countries.

Thanks for the suggestion. Relevant sections have been modified in this study, "Previous studies were shown that PAHs in the atmosphere of heavily polluted areas such as factories and the urban posed a threat to human health, especially the respiratory system (Agudelo-Castañeda et al., 2017; Ramírez et al., 2011)" see Line 51-53, "According to the statistics, the incidence and mortality of lung cancer were ranked first among cancer-related cases in the world, and so the lung cancer risk owing to exposing to PAHs was of particular concern and widely assessed (Křůmal and Mikuška,

2020; Liao et al., 2011; Taghvaee et al., 2018; Zhang et al., 2023)" see Line 56-59.

(3) Figure 1. abbreviations for PAHs. Abbreviations need to be defined in the main text before their first appearances.

Thanks for the suggestion. It has been defined in the figure notes, "Atmospheric concentrations of polycyclic aromatic hydrocarbons (PAHs) around the BS from June 2014 to May 2019" see Line 226.

(4) Line 207 What is the significant digit for the data in the study? Uncertainties of both chemical analysis and sampling (e.g., uncertainties in sampling flow rate, time) would need to be considered for providing numbers. The reviewer suggests the authors to revisit the significant digit for all the numbers (e.g., concentrations and fraction) in revising the manuscript.

The significant digit for the data of this study were modified according to the previous research results of our research group (Wang et al., 2018), and the numbers in this manuscript were modified, such as in Line 27, Line 213, Line 232-234, Line 254-255, and Line 279.

(5) L223 Why was the spring selected as the start of the cycle?

Because the sampling time of this study lasted from summer 2014 (June) to spring 2019 (May) for 5 years. In order to better characterize the change of PAHs, a whole year was selected as the research object, and the period from summer 2014 to spring 2015 was exactly one year, so summer was selected as the beginning of the cycle.

(6) L241 Could the authors provide some potential reasons why emission from high temperature combustion sources reduced during the study period?

High temperature combustion emission sources mainly included industrial and vehicle emissions, such as coal-fired power plants, automobile exhaust. The Chinese government has implemented various stringent measures to reduce air pollutant emissions over the past two decades, such as promoting ultra-low emission and ultra-high combustion technologies, promoting new energy vehicles and implementing strict emission standards, and strengthening process optimization and energy efficiency. Air pollutants from coal-fired power plants decrease from 2014 to 2019. (Wang et al., 2020) PAHs emissions from industrial coal, gasoline, and diesel oil increased by 7.8, 10.8, and 10.0 times, respectively, from 1980 to 2016, and then decreased by 4.3%, 13.5%, and 17.6%, respectively, during the period from 2017 to 2020. (Cao et al., 2022)

(7) L255 The meaning of the sentence is unclear. Please update.

Thank you for your advice. The sentence has been corrected by consideration of context.

The original text was "This indicated that there were other important pollution sources for LMW-PAHs, followed by MMW-PAHs, which was significantly increasing in winter at the BS region.". It was changed to the sentence "This indicated that there were nonnegligible pollution sources for LMW-PAHs, especially in winter at the BS region.", see Line 262-263.

(8) L259 The statement is supported by a reference in 2007. I believe that there must have been some changes in heating in China during the last 15 years. Is there a better reference that supports the statement?

Thank you for the suggestion. "For typical northern families, the consumption of firewood burning and coal in winter was 1.5−2.0 times higher than that in summer due to heating and other activities (Qin et al., 2007). As a result, PAHs emissions in winter were at least 1.5 times higher than those in summer." changed into "In terms of the per capita fuel consumption spatial distribution, the north and west China were apparently higher than that of southeast China, principally because of the difference in winter heating fuel consumption. Therefore, there were significant seasonal variations of per capita fuel consumption, with peak consumption in the winter months being about twice as high as in the summer months. (Zhu et al., 2013)", see Line 266-270.

(9) L280 L279 Back trajectory analysis is useful for estimating sources of air masses. However, it does not guarantee that measured atmospheric trace species are also transported through the path. A combined analysis with spatial distribution of emission sources is needed to support the statement.

Thank you very much for your advice. In the manuscript, we added the relevant spatial distribution information of pollution emission, see Line 290-293 "According to the distribution of atmospheric PAHs in some representative parts of northern China, it was found that the Beijing-Tianjin-Hebei region was greatly affected by nearby sources, while Shandong province and other places were mainly affected by regional emissions. (Zhang et al., 2016)".

[Figure]

**Fig. 1.** Measured ambient concentration and emission inventory of total PAHs in North China ("S" denotes site).

(10) L314 what does 'theoretical Q value' mean? Could the authors add a reference to support the statement?

Thank you for your advice. The theoretical Q value in the original sentence was the value of PMF model under ideal conditions. The theoretical Q value was the number of data input to the PMF model minus the number of data available for factor calculation. (Sun et al., 2021). And the reference was added, see Line 326.

**References**

Agudelo-Castañeda, D.M., Teixeira, E.C., Schneider, I.L., Lara, S.R., Silva, L.F.O.: Exposure to polycyclic aromatic hydrocarbons in atmospheric $PM_{1.0}$ of urban environments: Carcinogenic and mutagenic respiratory health risk by age groups, Environ. Pollut., 224, 158-170, https://doi.org/10.1016/j.envpol.2017.01.075, 2017.

Cao, X., Huo, S., Zhang, H., Zhao, X., Guo, W., He, Z., Ma, C., Zheng, J., Song, S.: Polycyclic Aromatic Hydrocarbons in China: Will Decoupling of Their Emissions and Socioeconomic Growth Emerge?, Earths. Future., 10, (1), https://doi.org/10.1029/2021EF002360, 2022.

Křůmal, K., Mikuška, P.: Mass concentrations and lung cancer risk assessment of PAHs bound to PM1 aerosol in six industrial, urban, and rural areas in the Czech Republic, Central Europe, Atmos. Pollut. Res., 11, 401-408, https://doi.org/10.1016/j.apr.2019.11.012, 2020.

Liao, C.M., Chio, C.P., Chen, W.Y., Ju, Y.R., Li, W.H., Cheng, Y.H., Liao, V.H.C., Chen, S.C., Ling, M.P.: Lung cancer risk in relation to traffic-related nano/ultrafine particlebound PAHs exposure: A preliminary probabilistic assessment, J. Hazard. Mate., 190, 150-158, https://doi.org/10.1016/j.jhazmat.2011.03.017, 2011.

Qin, F., Liu, H.Y., Jing, L., Liu, W., Yin, J., and Wei, X.D.: The investigation of energy con-sumption in the village of Jilin province, Journal of Jilin Jianzhu University (China)., 2, 37-40, https://d.wanfangdata.com.cn/periodical/jljzgcxyxb200702011, 2007.

Ramírez, N., Cuadras, A., Rovira, E., Marcé, R.M., Borrull, F.: Risk Assessment Related to Atmospheric Polycyclic Aromatic Hydrocarbons in Gas and Particle Phases near Industrial Sites, Environ, Health. Perspect., 119, 1110-1116, https://doi.org/10.1289/ehp.1002855, 2011.

Sun, Z., Zong, Z., Tian, C., Li, J., Sun, R., Ma, W., Li, T., Zhang, G.: Reapportioning the sources of secondary components of PM2.5: combined application of positive matrix factorization and isotopic evidence, Sci. Total. Environ., 764, https://doi.org/10.1016/j.scitotenv.2020.142925, 2021.

Taghvaee, S., Sowlat, M.H., Hassanvand, M.S., Yunesian, M., Naddafi, K., Sioutas, C.: Source-specific lung cancer risk assessment of ambient $PM_{2.5}$-bound polycyclic aromatic hydrocarbons (PAHs) in central Tehran, Environ. Int., 120, 321-332, https://doi.org/10.1016/j.envint.2018.08.003, 2018.

Wang, G., Deng, J., Zhang, Y., Zhang, Q., Duan, L., Jiang, J.; Hao, J.: Air pollutant emissions from coal-fired power plants in China over the past two decades, Sci. Total. Environ., 140326, 741, https://doi.org/10.1016/j.scitotenv.2020, 2020.

Wang, X., Zong, Z., Tian, C., Chen, Y., Luo, C., Tang, J., Li, J., Zhang, G.: Assessing on toxic potency of $PM_{2.5}$-bound polycyclic aromatic hydrocarbons at a national atmospheric background site in North China. Sci. Total. Environ., 612, 330-338, https://doi.org/10.1016/j.scitotenv.2017.08.208, 2018.

Zhang, X., Leng, S., Qiu, M., Ding, Y., Zhao, L., Ma, N., Sun, Y., Zheng, Z., Wang, S., Li, Y., Guo, X.: Chemical fingerprints and implicated cancer risks of Polycyclic aromatic hydrocarbons (PAHs) from fine particulate matter deposited in human lungs, Environ. Int., 173, https://doi.org/10.1016/j.envint.2023.107845, 2023.

Zhang, Y., Lin, Y., Cai, J., Liu, Y., Hong, L., Qin, M., Zhao, Y., Ma, J., Wang, X., Zhu, T., Qiu, X., Zheng, M.: Atmospheric PAHs in North China: Spatial distribution and

sources. Sci. Total. Environ., 565, 994-1000, https://doi.org/10.1016/j.scitotenv.2016.05.104, 2016.

Zhu, D., Tao, S., Wang, R., Shen, H., Huang, Y., Shen, G., Wang, B., Li, W., Zhang, Y., Chen, H., Chen, Y., Liu, J., Li, B., Wang, X., Liu, W.: Temporal and spatial trends of residential energy consumption and air pollutant emissions in China, Appl. Energ., 106, 17-24, https://doi.org/10.1016/j.apenergy.2013.01.040, 2013.

---

## Author Comment (AC2)

In this study, PAHs in atmosphere were analyzed from 2014 to 2019 in the Bohai Sea in China. It was found that the atmospheric concentrations were decreased in this period, and the composition of PAHs was also changed caused by the pollution prevention and control. Finally, the cost with the lung cancer caused by PAHs exposure was also estimated, and it was also found the decreasing trend from 2014 to 2019. In general, the study was well organized, and the data was enough for obtaining strong conclusion as the study mentioned. The study can be considered as a key area for the air pollution control in China, which also has scientific contributions to related field. The structure and English writing were easy for understanding. There are some suggestions for the revision.

(1) Some writings should be carefully checked, such as BS is the most polluted area of PAHs in China? Is there any supporting information?

Thank you very much for the advice. The Bohai Rim region was the third pole of China's economic development, with many economic development areas. The rapid population growth, urbanization, and industrialization led to increased pollution for the BS region. It was estimated that the PAHs emissions of the three provinces around the Bohai Sea Economic Circle (Liaoning, Hebei, and Shandong) accounted for 21% of the national emissions (Shen et al., 2013). And the emissions of the near four provinces (Shanxi, Henan, Anhui, and Jiangsu) accounted for 20% of the national emissions (Xu et al., 2006). Under the effect of the west wind, the outflow to the east accounted for 80% of the total outflow, and the emission source was mainly in North China (Zhang et al., 2011). In the process of atmospheric outflow, about 70% of PAHs deposited at the coastal zone and offshore area in the east of China (Lang et al., 2008). Therefore, the concentration of PAHs at the BS and its surrounding areas was at a relatively high level. Therefore, the original sentence of the manuscript was changed to "The Bohai Sea (BS) as one of the severe polluted areas of polycyclic aromatic hydrocarbons (PAHs) in China has been received wide attention in recent decades.", see Line 22-23 and "PAHs pollution in the atmosphere of the Bohai Sea was in a severe situation (Wang et al., 2018).", see Line 65-66.

(2) In the introduction section, the relationship between BTH and BS should be clearly mentioned. In my opinion, too many information was added for BTH, however, more information should be added for BS.

Thanks for the advice. The paper added the introduction of the relationship between the BTH region and the BS region. "The Bohai Rim economic area included the Liaodong Peninsula, the Shandong Peninsula, and the Beijing-Tianjin-Hebei (BTH) region. The BTH region was the center of economic development of the Bohai Rim economic area. (Liang et al., 2018; Zhang et al., 2016)", see Line 66-68.

(3) Line 125, what's the objective of adding hexamethylbenzene?

Hexamethylbenzene was used as an internal standard for the instrumental quantitative analysis.

(4) For QA and QC, the information for background was missing, which was important for PAHs analysis.

Each batch of samples contained one laboratory blank and one field blank. The pretreatment process of the blanks was consistent with the samples. No target of PAHs was detected in laboratory blanks. Acy, Ace, Flu, Phe, Fla, and Pyr were detected in field blanks, and the values of Acy, Ace, Flu, Phe, Fla, and Pyr were 0.04 - 0.10, 0.04 - 0.09, 0.03 - 0.08, 0.01 - 0.04, ND (Not detected) - 0.01, and ND (Not detected) - 0.02. Relevant information was added to the Table S2 of SI.

(5) For the calculation of ILCR, is the method widely applied in related studies? As I know, the USEPA has other method for ILCR with more parameters, such as body weight, breathing-rate.

Thank you very much for the suggestion. The USEPA has the method for ILCR with more parameters, such as body weight, breathing-rate. It could also be estimated by using Eq. (4), and it had been approved (Zhuo et al., 2017).

(6) Lin 213 to 218, if the cited studies did not use the passive air sampling method, the comparison cannot be made. Because there are some uncertainties between active and passive air sampling methods, which will influence the comparison conclusion.

Thank you very much for your advice. There was some uncertainty between active and passive air sampling methods. In this study, the sampling volume of the passive sampler

was taken into consideration when calculating the PAHs concentration of the sample, which has been reported in previous studies (Jaward et al., 2005; Moeckel et al., 2009).

(7) Line 300, for figure 2 the error bar should be added for average or mean values, the same problem should also be corrected for the whole manuscript.

The error bar in Figure 2 was added, see Line 310. Other relevant issues have been corrected, such as Figure 5 shown in Line 435.

(8) For Section 3.2, TJ was selected as the special site for BS, and the sources apportionments were both analyzed for TJ and BS. I just want to know, for the analysis of BS, the data of TJ was excluded from BS or not? If the data of TJ was included in BS for analysis, it may influence the results, please check.

Thank you for your advice. The data of TJ was not excluded from the BS. Since this study evaluated the source apportionment was analyzed for the entire BS, TJ was not separated from the BS. At the same time, TJ was an important PAHs emission source for the BS. (Zhang et al, 2016)

(9) Line 328, what's the type of fuel combustion?

Line 328 "Fuel combustion emissions were the reason for the significant increase of atmospheric pollutants, and that were also responsible for the elevated $PM_{2.5}$ levels around the BS region." in the sentence "Fuel combustion" referred to the combustion of fossil fuels, and it has been modified in the manuscript, see Line 339.

(10) Line 383-384, the related figure should be added, just like Figure 4.

Line 383-384 "Generally, the seasonal distribution of five sources for atmospheric PAHs at TJ was consistent with that the BS, which was not separately discussed here". The relevant information of TJ was shown as Figure S10 of SI, see Line 395-396.

(11) Line 411-412, the sentence was confused.

Thank you for your suggestion. "The decrease in the fifth year compared with the first year was more than two times, indicating that the environmental health risk of PAHs was decreasing" was modified to "The environmental health risk of PAHs in the fifth year was decreased by three times than in the first year", see Line 421-423.

(12) Line 436, the value of ILCR was higher than $10^{-4}$, which indicated life-threatening for human Same as Comment (5), for Section 3.3, the major concern is the suitability

of Eq. (4), please check.

Firstly, Line 436 meant to express the whole ILCR level of the Bohai Sea. Eq. (4) in the reference research (Zhuo et al., 2017), it referred to "Health risk due to inhalation exposure of PAHs was characterized by estimating the Incremental Lifetime Cancer Risk (ILCR) using a point-estimate approach". The sites set up in this study involve the whole Bohai Sea region and were evenly distributed. The evaluation of ILCR around the Bohai Sea was based on the overall evaluation of 12 sites atmospheric PAHs concentrations.

(13) Line 484, the two scenarios should be mentioned clearly here.

Thank you very much for the advice. Corrections have been made in the manuscript, the "Figure 6 shows the comparative results of direct medical costs of lung cancer at the BS region and TJ under two scenarios from 2014 to 2018" was changed into "Figure 6 shows the comparative results of direct medical costs of lung cancer at the BS region and TJ from 2014 to 2018 under before and after pollution control", see Line 493-494.

(14) Line 488, five times? Please check.

Thank you for your advice. Modifications have been made see Line 492 "Assuming that no air pollution control was implemented, the total direct medical costs of lung cancer caused by PAHs exposure did not change in five years, and the total direct medical costs was $23.2 million.", see Line 498.

**References**

Liang, X., Tian, C., Zong, Z., Wang, X., Jiang, W., Chen, Y., Ma, J., Luo, Y., Li, J., Zhang, G.: Flux and source-sink relationship of heavy metals and arsenic in the Bohai Sea, China, Environ. Pollut., 242, 1353-1361, https://doi.org/10.1016/j.envpol.2018.08.011, 2018.

Zhang, Y.J., Lin, Y., Cai, J., Liu, Y., Hong, L.N., Qin, M.M., Zhao, Y.F., Ma, J., Wang, X.S., Zhu, T., Qiu, X.H., and Zheng, M.: Atmospheric PAHs in North China: Spatial distribution and sources, Sci. Total. Environ., 565, 994 1000, https://doi.org/10.1016/j.scitotenv.2016.05.104, 2016.

Shen, H., Huang, Y., Wang, R., Zhu, D., Li, W., Shen, G., Wang, B., Zhang, Y., Chen,

Y., Lu, Y., Chen, H., Li, T., Sun, K., Li, B., Liu, W., Liu, J., Tao, S.: Global Atmospheric Emissions of Polycyclic Aromatic Hydrocarbons from 1960 to 2008 and Future Predictions, Environ. Sci. Technol., 47, 6415-6424, https://doi.org/10.1021/es400857z, 2013.

Xu, S.S., Liu, W.X., Tao, S.: Emission of polycyclic aromatic hydrocarbons in China, Environ. Sci. Technol., 40, 702-708, https://doi.org/10.1021/es0517062, 2006.

Zhang, Y., Shen, H., Tao, S., Ma, J.: Modeling the atmospheric transport and outflow of polycyclic aromatic hydrocarbons emitted from China, Atmos. Environ., 45, 2820-2827, https://doi.org/10.1016/j.atmosenv.2011.03.006, 2011.

Lang, C., Tao, S., Liu, W., Zhang, Y., Simonich, S.: Atmospheric transport and outflow of polycyclic aromatic hydrocarbons from China, Environ. Sci. Technol., 42, 5196-5201, https://doi.org/10.1021/es800453n, 2008.

Zhuo, S., Shen, G., Zhu, Y., Du, W., Pan, X., Li, T., Han, Y., Li, B., Liu, J., Cheng, H., Xing, B., and Tao, S.: Source-oriented risk assessment of inhalation exposure to ambient polycyclic aro-matic hydrocarbons and contributions of non-priority isomers in urban Nanjing, a megacity lo-cated in Yangtze River Delta, China, Environ. Pollut., 224, 796   809, https://doi.org/10.1016/j.envpol.2017.01.039, 2017.

Jaward, T.M., Zhang, G., Nam, J.J., Sweetman, A.J., Obbard, J.P., Kobara, Y., and Jones, K.C.: Passive air sampling of polychlorinated biphenyls, organochlorine compounds, and polybrominated diphenyl ethers across Asia, Environ. Sci. Technol., 39, 8638–8645, https://doi.org/10.1021/es051382h, 2005.

Moeckel, C., Harner, T., Nizzetto, L., Strandberg, B., Lindroth, A., and Jones, K.C.: Use of dep-uration compounds in passive air samplers: results from active sampling-supported field de-ployment, potential uses, and recommendations, Environ. Sci. Technol., 43, 3227   3232, https://doi.org/10.1021/es802897x, 2009.